# Using Micropropagation to Develop Medicinal Plants into Crops

**DOI:** 10.3390/molecules26061752

**Published:** 2021-03-21

**Authors:** Rita M. Moraes, Antonio Luiz Cerdeira, Miriam V. Lourenço

**Affiliations:** 1Santa Martha Agro Ltd.a, Rodovia Prefeito Antonio Duarte Nogueira, Km 317, Contorno Sul, Ribeirão Preto, SP 14.032-800, Brazil; mvlouren@gmail.com; 2Fundação Fernando E. Lee, Av. Atlântica 900, Balneário, Guarujá, SP 114420-070, Brazil; 3Embrapa Meio Ambiente, Rodovia SP-340, Km 127,5, Tanquinho Velho, Jaguariúna, SP 13918-110, Brazil; antonio.cerdeira@embrapa.br

**Keywords:** medicinal plants, in vitro propagation, medicinal crops, phytomedicines

## Abstract

Medicinal plants are still the major source of therapies for several illnesses and only part of the herbal products originates from cultivated biomass. Wild harvests represent the major supply for therapies, and such practices threaten species diversity as well as the quality and safety of the final products. This work intends to show the relevance of developing medicinal plants into crops and the use of micropropagation as technique to mass produce high-demand biomass, thus solving the supply issues of therapeutic natural substances. Herein, the review includes examples of in vitro procedures and their role in the crop development of pharmaceuticals, phytomedicinals, and functional foods. Additionally, it describes the production of high-yielding genotypes, uniform clones from highly heterozygous plants, and the identification of elite phenotypes using bioassays as a selection tool. Finally, we explore the significance of micropropagation techniques for the following: a) pharmaceutical crops for production of small therapeutic molecules (STM), b) phytomedicinal crops for production of standardized therapeutic natural products, and c) the micropropagation of plants for the production of large therapeutic molecules (LTM) including fructooligosaccharides classified as prebiotic and functional food crops.

## 1. Introduction

For over a century, plant tissue culture technology has been an important tool in crop improvement and development: producing disease-free plant material [1], obtaining transgenic plants [2,3], breaking dormancy, and micropropagating elite plants with highly desirable chemotype [4], thereby leading to more uniform plant production [5,6]. This is the technology for conserving in vitro germplasm of elite [7,8], rare, and endangered plant species [9,10,11], implementing breeding programs for innumerous crops as well as encapsulated seeds [12], and studying plant biosynthesis through cell and root cultures [12,13], the interaction between endophytes and the hostplant [14,15].

High-demand plants face great challenges: Depletion of species diversity due to overharvesting and environmental pollution affecting natural populations are strong factors that support the argument for cultivating rare and elite high-yielding medicinal plants. In addition, the cultivation of medicinal plants is the most effective way of addressing the gap between supply and demand. Breeding studies are necessary both to develop pharmaceutical plants as crops and to scale up their production [16]. Still, few success stories about breeding medicinal plants such as *Artemisia annua* L. exist. Because micropropagation is the tool of producing clones—especially with high-yielding chemotypes—for industrial purposes, it solves this target-breeding problem. Moreover, as the *Echinacea* study [17] showed, micropropagation’s demonstrated ability for mass selection suggests that together with bioassays it could form part of an overall strategy to screen elite phenotype lines.

Micropropagation is an in vitro technology of rapidly multiplying elite plants using modern plant tissue culture methods. It is well-known for its applications in the agro, horticultural and forestry industries, this review focuses on a less-commonly known area which is on medicinal plants and the need to develop them as medicinal crops. Li et al. [18] defined pharmaceutical crops in three distinct categories: 1) crops for the production of small therapeutic molecules (STMs), 2) standardized therapeutic extracts (STEs), and 3) large therapeutic molecules (LTMs). In addition, this review also examines micropropagation of functional food plants to ensure their development as crop.

## 2. Pharmaceutical Crops for Production of Small Therapeutic Molecules (STM) 

Drug discovery programs and the formation of knowledge of different pharmacological classes of pharmaceuticals owe much to traditional medicine in countries such as China and India [19,20]. Some natural compounds are extracted or used as templates for synthesis or as a precursor for the semi-synthesis (e.g., paclitaxel, artemisinin, podophyllotoxin, cannabinoids, galantamine, vinca alkaloids, atropine, ephedrine, digoxin, morphine, quinine, reserpine, tubocurarine etc.). Many of these compounds provide therapeutic relief for several major illnesses including cancer, Alzheimer, malaria, high blood pressure, fever, and anxiety. As researchers confirm the medicinal utility of these natural resources, they suffer depletions with the increased demand. 

According to McChesney et al. [20], pharmaceutical natural substances require considerations beyond supply and demand: the establishment of successful production systems must be sustainable, environmentally safe, reliable, and affordable. Thus, the development of medicinal crops is a key factor to obtaining a commercially viable source of medicinal biomass for the pharmaceutical industry. In fact, non-stable supply sources could lead to bottlenecks that limit potentially beneficial products. For example, researchers pointed to insufficiency in the biomass supply of anti-cancer pharmaceutical ingredients such as podophyllotoxin and paclitaxel, as the major limiting factor at phase III clinical trials, which led to overharvesting of the natural resources of *Podophyllum emodi* Wall ex Royle in India [21] and *Taxus baccata* L. in Europe [22].

Given the shortage of biomass supply limiting clinical phase III trials of paclitaxel and podophyllotoxin, several laboratories engaged in different research approaches that included bioprospecting studies searching for alternate sources with high yields of the active compounds [23,24,25]. Clippings of cultivated *Taxus* sp. became the reliable source for production of paclitaxel [18], and Sisti et al. [26] reported methods of semi-synthesis using abundant intermediates for production of paclitaxel. Majada et al. [27] reported a procedure to obtain high-yielding *T. baccata* plantlets by screening micropropagated juvenile seedlings that accumulate 10-deacetyl baccatin III. The selected genotypes of *T. baccata* grow faster and contain high taxene content.

For its part, podophyllotoxin is the starting compound for semisynthesis of etoposide and teniposide, two potent DNA topoisomerase cancer drugs utilized in the treatment of small lung and testicular cancers, lymphomas/leukemias and the water-soluble etoposide phosphate, also known as etopophos (Figure 1). To assure podophyllotoxin supply, a buffer extraction procedure using leaf biomass of mayapple plants provides a sustainable alternative source [28]. Later, we published a survey and a database of high-yielding podophyllotoxin colonies [29,30] and an in vitro propagation protocol of **Podophyllum peltatum** L. to rapidly produce podophyllotoxin-rich plantlets [5].

*Artemisia annua* L. is the source of artemisinin, an endoperoxide sesquiterpene lactone that is very difficult to synthesize, precursor of a common anti-malarial drug (Artemether). Artemisinin production comes from cultivated plants selected for their high artemisinin content [18]. Selection of genotypes with high artemisinin concentration in wild populations resulted in lines containing up to 1.4 percent artemisinin (on dry leaves basis). The leading commercial source, ‘Artemis,’ exhibited extensive variation of metabolic and agronomic traits; artemisinin content on a µg/mg dry basis for individual plants ranged 22-fold, plant fresh weight varied 28-fold, and leaf area ranged 9-fold [31].

While Ferreira and Janick [32] found that the in vitro production of artemisinin will never be commercially feasible, Wetzetein et al. [33] suggested that cultivation of micropropagated high-yielding artemisinin plants with levels above 2% and improved agronomic traits (high leaf area and shoot biomass production) may reach productivity of 70 kg/ha artemisinin using a crop density of 1 plant m^−2^. We include in Table 1 examples of pharmaceutical plant species classified as small therapeutic molecules STM’s (18) and their micropropagation protocols to produce elite clones for higher yields. Taxol^®^ is another success story. According to McChesney et al. [20], the path from the discovery to a pharmaceutical drug requires a viable production system (cultivation, harvest, extraction, purification and isolation) where every step of a natural product must be systematically evaluated. Micropropagation of the superior source (chemotype or variety of the species) may help to produce biomass with a high and consistent concentration of the natural product or a precursor of the natural product that can be converted economically by semi-synthesis to the final bulk active product.

## 3. Phytomedicine Crops for the Production of Standardized Therapeutic Natural Products 

Herein, we describe the category of phytomedicinal crops similar to what Li et al. [18] reports regarding pharmaceutical crops for production of standardized therapeutic extracts (STEs). Additionally, we relate examples wherein micropropagation proves useful as a method for ensuring the stability of biomass supply of phytomedicines by allowing breeders to select phytomedicinal crops with an eye towards maintaining genetic consistency and the sustainability of wild plant population.

Also known as botanical drugs, herbal remedies, and herbal medicines, phytomedicines are classified in the United States as dietary supplements according to the specific claim as described in the Dietary Supplement Health and Education Act (DSHEA) of 1994 [18]. In Europe, the phytomedicines are standardized and certified medicinal products and in Asia they have a status of traditional medicine. Phytomedicinal crops relate to the cultivation of medicinal species by which a mixture of multiple active compounds commercialized as standardized products. Usually, phytomedicines are evaluated for quality as the means to ensure safety, as complex mixtures of secondary compounds, to maintain consistency is fundamental to their efficacy. Thus, authenticity and uniformity and well-defined cultivation practices and postharvest processes are essential to certify safety and efficacy. Govidaraghavan and Sucher [44] reinforce that herbal productions must follow good agricultural and collection practice (GACP), good plant authentication and identification practice (GPAIP), good manufacturing practice (GMP) before and during the manufacturing process, guided by analytical tools, and micropropagation is an important tool in ensuring uniformity and consistency in open pollinated crops.

As of today, the majority of phytomedicines are still harvested from the wild, which causes habitat destruction, genetic diversity loss, as well as ingredient mislabeling, variability and contamination. In Brazil, products are sourced from the wild, as well as from cultivation in agroforest or in small gardens. They are chosen without proper guide from health-care professionals because medical schools do not include in their curriculum the disciplines of phytomedicines or phytotherapy. In 2016 the Brazilian Health Regulatory Agency, ANVISA, officially recognized twenty-eight medicinal plant species as herbal drugs and published their monographs [45] in the first edition (Memento). The monographs are a complete therapeutic guide of phytomedicinals reviewed and accepted by ANVISA as therapies used in SUS, the public health system of Brazil. The majority of the phytomedicinals included in this first edition, was introduced to Brazil by immigrants and later became part of traditional use especially by the rural communities.

The increased consumption of phytomedicine offers an opportunity to develop medicinal plant production systems as crop. Conventional plant breeding may improve agronomic traits in association with molecular markers aiding crop development. The greatest obstacles for such a program remain predicting which extracts remain active, specifically resembling all the medicinal properties described in the ones harvested in the wild [46]. In this context, micropropagation may produce clones that could be screened using bioassays to assure bioactivity. Moraes et al. [17] used tissue culture techniques to produce *Echinacea* sp. clones and later screened those using human monocytes assays to identify high and low activity. The immune response between the two selected clones after field cultivation due to bacterial endophytes was the same [47]. The selection procedure using in vitro propagation techniques, genetic markers, and bioassay work are approaches for selection of elite germplasm [17].

Micropropagation allows one to mass generate plants with genetically identical chemotype for cultivation purposes. Reinhard [48] suggested that different chemotypes in Cat’s Claw (*Uncaria tometosa* (Willd. ex Schults) DC) might have different healing properties: tetracyclic oxindole alkaloid acting on the central nervous system, and the pentacyclic oxindole alkaloid affecting the immune system. The immunological effect of both alkaloid mixtures is antagonistic and therefore may be unsuitable for therapy. For Reinhard [48], the production of safe and efficacious Cat’s Claw phytomedicinal requires chemical identification prior to harvesting and perhaps even before the cultivation.

Micropropagation also allows one to select plants based on the chemical profile in order to standardize a particular chemotype. Morais et al. [49] reported that the chemical composition of *Lippia sidoides* Cham. (syn. *Lippia origanoides)* varied according to cultivation sites. Thymol is the major component of essential oil extracted from crops grown in northeast Brazil [50,51,52], whereas carvacrol is the major component present in *L. sidoides* harvested from Lavras, Minas Gerais [53] and 1.8-cineole, isoborneol, and bornyl acetate in São Gonçalo do Abaeté, Minas Gerais, Brazil. Standardized essential oil of *L. sidoides* is recommended for topical applications on skin, mucous membranes, mouth, throat and vaginal washings as antiseptic [45]. According to Santos et al. [53], genotypes regulate chemical polymorphism thymol and carvacrol. Phenotypical variation is likely to influence biological properties and the type of industrial application. Planting thymol or carvacrol clones ensured a high-quality biomass for safe and efficacious products [54].

Finally, micropropagation proves useful to reduce consumption pressure on potentially threatened wild populations [55]. For example, bark extraction of barbatimão to produce phytomedicine has depleted genetic diversity of *Stryphnodendron polyphythum* Mart. natural resources. The bark of this Brazilian tree is widely utilized as a wound-healing phytomedicine with anti-inflammatory, antioxidant and antimicrobial activities. Souza-Moreira et al. [55] showed that proanthocyanidins present in the bark are responsible for its healing properties. França et al. [10] published an efficient micropropagation protocol to produce barbatimão plantlets, while Correa et al. [56] defined the conditions for in vitro germplasm conservation to reduce pressure on its threatened status. Table 2 includes in vitro propagation protocols to produce healthy plantlets for cultivation purposes, thus aiding the development of phytomedicinal crops.

As the above paragraphs state, micropropagation can provide an effective technique to those seeking to mold a supply chain of a product, in order to ensure the genetic homogeneity of plant clones, chemical profile, and finally sustainability of those plants harvested in the wild. 

## 4. Micropropagation of Plants for Production of Large Therapeutic Molecules (LTM) Including Fructooligosaccharides Classified as Prebiotic

Li et al. [18] has called on LTMs crop plants to be cultivated for production of large molecules such as proteins and polysaccharides and engineered crops (GM) with the ultimate goal of producing drugs or vaccines at low cost. The LTM’s crops are sources of proteolytic enzymes such as papain isolated from *Carica papaya* L., bromelain from fruits and stems of pineapple, and the bioactive momordica anti-HIV from *Momordica charantia* L. We included in the LTM’s species those that supply prebiotic dietary fibers that are carbon sources for fermentation pathways in the colon to support digestive health. In this section, we focus on these prebiotic fibers to highlight how micropropagation may be used to create a stable supply of crops that produce LTMs. 

Fructooligosaccharides/inulin also known as FOS are universally agreed-upon prebiotics [77], and species that are rich sources of dietary fibers have tremendous effect on gut microbiome. Humans cannot digest FOS. Instead, the gut microbiome ferments these non-digested carbohydrates and produces short chain fatty acids with health benefits such as reducing the risk of cancer and increasing the absorption of both calcium and magnesium. Research on the gut microbiome has increased exponentially, revealing that the intestines greatly affect human health, especially in relation to the immune system and behavior [78,79].

FOS are present in fruits, bulbs, rhizomes, and roots of banana, onion, garlic, and species belonging to the Agavaceae and Asteraceae, which are the richest sources of FOS including chicory (*Chicorium intybus* L.), globe artichoke (*Cynara cardunculus* var. *scolymus* L. Fiori, Jerusalem artichoke (*Helianthus tuberosus* L), elecampane *(Inula helenium* L.), bear’s foot (*Smallanthus uvedalia* (L.) Mack. ex Mack and yacon (*Smallanthus sonchifolius* (Poepp.) H. Rob.). According to Roberfroid [78] chicory roots provide the commercial source of FOS for industrial applications, also known as inulin, which are extracted and then processed into short-chain fructans, such as the oligofructose with 2–10 degree of polymerization by partial enzymatic hydrolysis. López and Urías-Silvas [79] reviewed the use of Agave/FOS as prebiotics called agavins whose molecular structure is composed of a complex mixture of fructans. The agavins stimulated the growth of *Bifidobacterium breve* and *Lactobacillus casei* more efficiently than most commercial inulin [80]. Melilli et al. [81] evaluated (*Cyanara cardunculus* var. *scolymus* L.) germplasm for inulin with a high degree of polymerization in the Mediterranean environment, to reduce breeding time and offer growers uniform, healthy globe artichoke plants. Ozsan and Onus [82] compared in vitro micropropagation response of open-pollinated cultivars with F1 hybrids in maturity and height. They concluded that open-pollinated cultivars are cheaper than F1 and could be used for in vitro mass propagation.

The food industry considers FOS/inulin a natural ingredient that improves sensory characteristics such as taste and texture, the stability of foams, emulsions and mouthfeel in a large range of food applications like dairy products and baked goods reducing sugar and fat content while improving health [83]. Padalino et al. [84] added inulin with different degree polymerization with whole meal flour to improve quality of functional wheat spaghetti as example of processed food. The Global Market Insights reported that inulin’s (FOS) market size in 2015 was 250 kilo tons and it is expected gains of 8.5% for 2023, likely to be worth more than US$ 2.5 billion [85]. The consumers of FOS are Europe, China, Japan and North America, with Japan being the world’s largest market. The COVID-19 pandemic reinforced the major role of microbiota on the immune response and well-being. We expect that more consumers will pay more attention to prebiotics that modulate the gut microbiome.

Recent studies on the traditional food yacon (*Smallanthus sonchifolius* (Poeppig & Endlicher) H. Robinson), an Andean species, demonstrated that its roots are also a rich source of FOS with a smaller degree of polymerization than chicory. It has great potential as a prebiotic and sugar substitute due to its sweet taste that is related to degree of polymerization [86,87]. The role of yacon as FOS supplementation favors a healthy microbiota while reducing pathogenic population in the gut. Furthermore, short chain fatty acids produced by the beneficial bacteria improve glucose homeostasis and lipid metabolism. Clinical studies confirm that consumption of yacon as flour or syrup prevented and treated chronic diseases [88,89]. The beneficial compounds present in storage roots of yacon classify the spices as functional food (Figure 2).

Brazil is one of the largest agricultural producers in the world, but does not produce inulin/FOS from either source (chicory or artichoke) for applications in the food industry. However, in the vicinity of Sao Paulo City, yacon is produced for fresh consumption for its health benefits. Thus, to initiate any production system for supplying FOS as an ingredient with applications in the food and pharmaceutical industries, rhizospheres [90], along with storage roots may be better utilized to extract FOS. Micropropagation of yacon can still be done using axillary buds as explants of healthy plantlets for cultivation. Table 3 shows the published in vitro protocols of FOS producing plants for development of the business models of fructans as prebiotic.

Given the predicted increase in FOS/inulin consumption, supply of these LTM’s crop plants will be necessary in a way such as the one suggested by McChesney et al. [20] a sustainable system to meet the demand. Yacon micropropagation is an example to stabilize the supply of crop plants as source of LTMs, thus ensuring that stability of production. 

## 5. Functional Food Crops

Metabolic syndrome is a global economic and social burden, understanding the origins, relevant factors contributing to high rates of obesity and its physiological impacts may reveal potential therapeutic targets. 

Maintaining a healthy gut microbiome is one of the therapeutic goals that improve human health [83]. Dietary fibers promote wellbeing, and thus they are classified as functional food. Wildman [96] refers as functional food, the food, either natural or formulated, which will enhance physiological performance or prevent or treat disease and disorders. 

Royston and Tolesfbol [97] refer to term epigenetic diet class of bioactive dietary compounds such as resveratrol in grapes, genistein in soybean, apigenin in celery, allicin in garlic, phenolic compounds in berries and omega 3 in **Portulaca oleracea** L. also known as purslanen [98] and other consumed foods, which have been shown to defend against the development of many different types of tumors. Compounds that act as epigenetic modulators prevent initiation and the progression of oncogenesis [97]. Micropropagation is an important tool for the propagation of selected lines in various breeding programs, as well as the recovery of pathogen-free material, or even for slow growth storage and the cryopreservation of valuable germplasm of fruit and vegetable crops.

## 6. Conclusions

Humans have long used plants to address various problems, the solutions to which often brought unintended consequences, such as overharvesting and environmental degradation. These negative consequences teach us the solution to our problems ought to be sustainable. Through a literature review, this paper argues that micropropagation can be a part of a strategy to reinforce the supply and quality of crops used for medicinal purposes: (1) small therapeutic molecules, (2) standard therapeutic extracts, (3) large therapeutic molecules, and (4) functional foods.

## Figures and Tables

**Figure 1 molecules-26-01752-f001:**
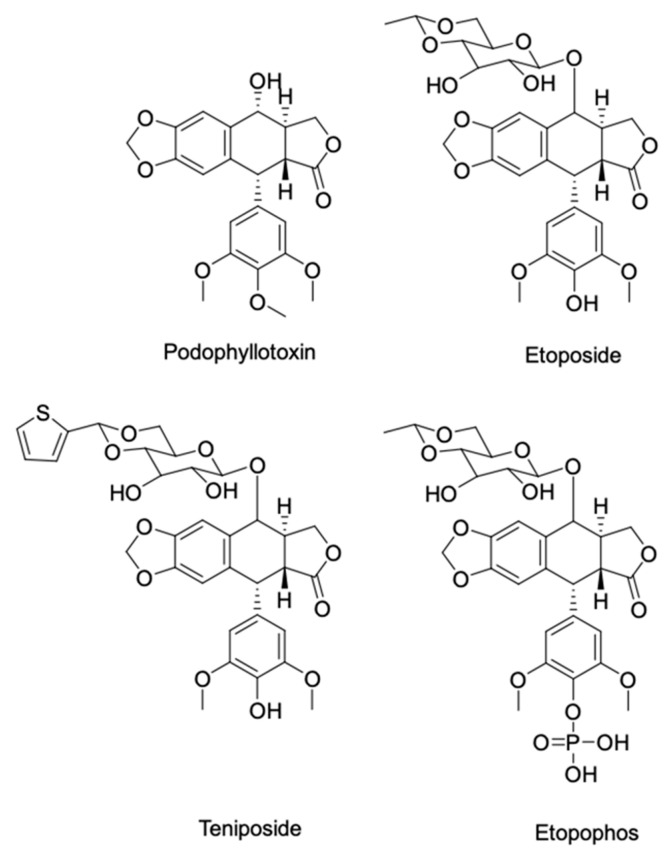
Structures of (−)-podohyllotoxin present in *Podophyllum* sp and its commercial chemotherapeutic derivatives.

**Figure 2 molecules-26-01752-f002:**
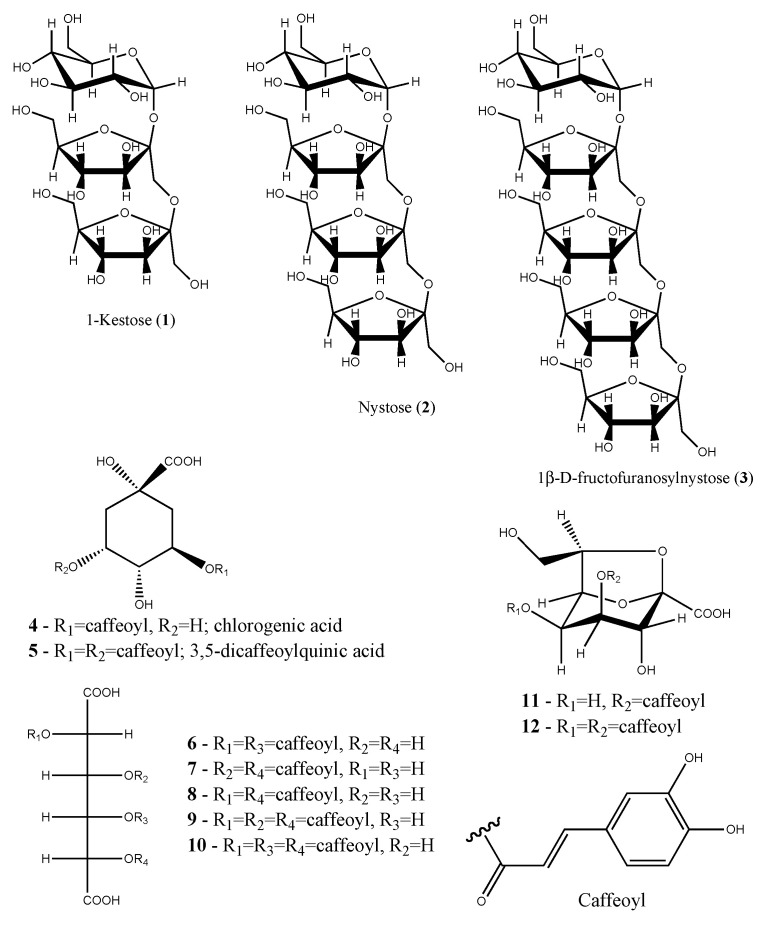
Yacon (*Smallanthus sonchifolius*) functional constituents.

**Table 1 molecules-26-01752-t001:** Commercial sources of pharmaceuticals often used in therapies of several illnesses that are micropropagated [17].

Plant Species	Natural Substance	Therapy	Micropropagation Protocol
*Artemisia annua* L.	Artemisinin	Antimalarial	Etienne et al. [34]
*Catharanthus roseus* (L.) G. Don	Vincristine, Vinblastine	Anticancer	Kumar et al. [35]
*Campotheca acuminata* Decne	Camptothecin	Anticancer, antiviral	Nacheva et al. [36]
*Leucojum aestivum* L.	Galantamine	Anti-alzheimer	Zagorska et al. [37]
*Narcissus* sp. L.	Galantamine	Anti-alzheimer	Khonakdari et al. [38]
*Hyoscyamus niger* L.	Scopolamine	Parasympatholytic	Uranbey et al. [39]
*Pilocarpus* sp. Vahl	Pilocarpine	Anti-glaucoma	Saba et al. [40]
*Podophyllum emodi* Wall ex Royle	Podophyllotoxin	Anticancer, antiviral	Chakraborty et al. [41]
**Podophyllum peltatum** L.	Podophyllotoxin	Anticancer, antiviral	Moraes-Cerdeira et al. [5]
*Rauwolfia serpentina* (L.) Benth ex Kurz	Reserpine	Hypotensive, sedative	Bhatt et al. [42]
*Taxus* sp L.	Paclitaxel	Anticancer	Abbasin et al. [43]

**Table 2 molecules-26-01752-t002:** Micropropagation protocols of medicinal plants considered phytomedicine by the Brazilian Regulatory Agency (ANVISA).

Plant Species (Common Name)	Herbal Constituents	Therapy	Micropropation Protocol
*Actaea racemosa* L. (Black cohosh)	Triterpenes	Hot flashes menopause	Lata et al. [57]
*Aesculus hippocastanum* L. (Horse chestnut)	Coumarins (Aesculetin), Triterpenoid Saponin Glycoside	Varicose vein syndrome	Sediva et al. [58]
*Allium sativum* L. (Garlic)	Thiosulfinates (Allicin), Terpenes	Bronchitis, asthma, arteriosclerosis	Ayabe and Sumi [59]
*Aloe vera* (L.) Burm. f. (Aloe)	Polysaccharides	Laxative, healing burns and wounds	Roy and Sarka [60]
*Calendula officinalis* L. (Calendula)	Flavonoids, Terpenes,	Anti-inflammatory, healing wounds	Çöçü et al. [61]
*Cynara scolymus* L. (Artichoke)	Flavonoids, Caffeoylquinic Acids	Hepatic-biliary, dysfunction and digestive complaints	El Boullani et al. [62]
*Echinacea purpurea* (L.) Moench (Echinacea)	Alkamides, Cichoric Acid, Polysaccharides	Cold treatment	Jones et al. [63]
*Ginkgo biloba* L. (Ginkgo)	Flavonoids, Terpene lactones	Circulatory disorders	Camper et al. [64]
*Harpagophytum procumbens* D*C.* ex Meisn. (Devil’s claw)	Iridoid glycosides	Anti-inflammatory	Kaliamoorthy et al. [65]
*Hypericum perforatum* L. (St. John’s wort)	Naphthodianthrones (Hypericin, pseudohypericin)	Antidepressant	Gadzovska et al. [66]
*Lippia sidoides* Cham. (Pepper rosmarin)	Essential Oils	Anti-inflammatory, antifungal, antiseptic	Costa et al. [54]
*Matricaria chamomilla* L. (Camomile)	Flavonoids, Essential Oils	antispasmodic, anti-inflammatory	Taniguchi & Tanakano [67]
*Maytenus ilicifolia* Mart. (Espinheira santa)	Flavonoids, Triterpenes	Gastric disordes	Pereira et al. [68]
*Passiflora incarnata* L. (Passion flower)	Flavonoids, Coumarin, Umbelliferone, Indol Alkaloids	Anxiolytic	Ozarowski &Thiem [69]
*Paullinia cupana* Kunth (Guaraná)	Caffeine	CNS stimmulant, antioxidant	Barbosa & Mendes [70]
*Peumus boldus* Molina (Boldo)	Essential oils, Aporphine Alkaloid, Flavonoids	Hepatic, diuretic, laxative	Rios et al. [71]
*Piper methysticum* G. Forst (Kava-kava)	Kavalactones	CNS activity, antidepression, anxiolytic	Zhang et al. [72]
*Psidium guajava* L. (Guava)	Tannins, Flavonoids, Triterpenes	Noninfectious diarrhea	Rawls et al. [73]
*Stryphnodendron adstringens (*Mart.) Coville (Barbatimão)	Tannins	Wound healing	França et al. [10]
*Uncaria tomentosa* (Willd. ex Schults) DC. (Cat’s claw)	Flavonoids, Alkaloids, Saponins, Triterpenes	Anti-inflammatory	Pereira et al. [74]
*Valeriana officinailis* L. (Valeriana)	Terpenes, Valepotriates, Lignans	Anxiolytic, insomnia, sedative	Abdi et al. [75]
*Zingiber officinale* Roscoe (Ginger)	Essential oils, Shogaol, Zingerone, Gingerol	Anti-inflammatory, anti-emetic and chemo-protective	Abbas et al. [76]

**Table 3 molecules-26-01752-t003:** Micropropagation protocols of FOS producing species.

Plant Species	Common Name	Culture Purposes	Microprogation Protocol
*Agave* sp L.	Agave, maguey	Production of high yielding plants	Robert et al. [91]
*Chicorium intybus* L. *Cuanara cardunculus var. scolymus* L.	Chicory Globe artichoke	Germplasm conservation, Improve root quality for medicinal value Propagation of open-pollinated cultivars	Previati et al. [92] Dolinski and Olek [93] Ozsan and Onus [82]
*Helianthus tuberosus* L.	Jerusalem artichoke	Large scale production of health plantlets	Abdalla [94]
*Smallanthus sonchifolius* (Poeppig & Endlicher) H. Robinson	Yacon	Production of healthy plantlets	Viehmannova et al. [95]

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
