# Peer review of "Using Micropropagation to Develop Medicinal Plants into Crops"

_molecules, 2021, doi:10.3390/molecules26061752_

Round 1

Reviewer 1 Report

This article is an interesting compilation on the potential modalities for unlimited supply of herbal constituents for therapeutic purposes. But the information provided is overclaiming and beyond current applications and hypothetical.

The title “Using Micropropagation to Develop Pharmaceuticals, Phyto-medicines and Functional Food Crops” – is itself misleading. Because, except for traditional medicines, the current herbal medicines are not direct utilization of plants. Lot of processing and purification is performed. In several established plant compound-based therapies, the molecule is actually an organic synthesized rather than directly from plants. So, the title should be reflecting this reality. To some extent, the claim on direct utilization of plants for functional foods is acceptable.

This reality also needs to be reflecting throughout the article. For example., line numbers 61 to 63.

Similarly, the section on Phytomedicine Crops for the Production of Standardized Therapeutic Natural (STN) Products. The source article which the authors have referenced does not provide any large detail regarding the production of STN’s, except for listing its salient features. Even that list clearly states that maintaining quality and induction are difficult and says that those plants are used only in the traditional medicine. But, the current article looks to be giving a different picture of the STN’s. It is quite understandable that many times phytochemical secondary metabolites quantity and concentration is difficult to be decided in the plants.  Several factors contribute to the production and in many cases the active extract also may not be a single compound.

Though this article is a good compilation of the micro-propagated plant resources for therapeutically active phytochemicals and functional foods, but the authors need to re-look on the content and need to tone down the content for not exaggerating the current scenario with reality.

Author Response

Reviewer 1:

Line 2: The title “Using Micropropagation to Develop Pharmaceuticals, Phyto-medicines and Functional Food Crops” was changed to “Using Micropropagation to Develop Medicinal Plants into Crops”. Highlighted.

Line 91 to 96: Tone down the content and importance of micropropagation. We have included a paragraph citing Dr. McChesney’s report the topic “Taxol® is another success story. According to McChesney et al [19] from the discovery to a pharmaceutical drug, a viable production system that includes cultivation, harvest, extraction, purification and isolation is required and every steps of production of a natural product must be systematically evaluated. Micropropagation of the superior source (chemotype or variety of the species) may help to produce biomass with a high and consistent concentration of the natural product or a precursor of the natural product that can be converted economically by semi-synthesis to the final bulk active product.” Highlighted.

Line 105 to 114: Phytomedicine crops for production of STN products, the authors of each source were referenced. In addition we included the importance quality control to ensure safety. “In the Europe, the phytomedicines are standardized and certified medicinal products and in Asia they have a status of traditional medicine. Phytomedicinal crops relate to the cultivation of medicinal species by which a mixture of multiple active compounds is commercialized as standardized products. Usually, phytomedicines are evaluated for quality as the means to ensure safety, as complex mixtures of secondary compounds, to maintain consistency is fundamental to their efficacy. Thus, authenticity and uniformity and well defined cultivation practices and postharvest processes are essential to certify safety and efficacy. Govidaraghavan and Sucher [44], reinforce that herbal productions must follow good agricultural and collection practice (GACP), good plant authentication and identification practice (GPAIP), good manufacturing practice (GMP) before and during the manufacturing process, and guided by analytical tools, and micropropagation is an important tool ensuring uniformity and consistency in open pollinated crops”. Highlighted.

Reviewer 2 Report

The review  Using Micropropagation to Develop Pharmaceuticals, Phyto-2 medicines and Functional Food Crops wants to  show the relevance of developing medicinal plants into  crops and the use of micropropagation as technique to mass produce high-demand biomass thus 15 solving supplies issues of therapeutic natural substances. The authors divided the paper in relation to the drug to valorise, but in each paragraph the advantage of micropropagation technique is reported. I suggest to introduce the technique at the beginning of the introduction and reduce the followed paragraph, otherwise it is repetitive.

In some crops, e.g. Cynara the literature about micropropagation is quite old and in other case other important crops for medicinal purposes are not mentioned, e.g. P oleracea.

Considering that the paper is a review and the authors report plant for functional food development I suggest to insert some new articles about developed functional foods with drug addition in relation to the plant considered.

The following paper should be add for instance:

Efficient plant regeneration and Agrobacterium-mediated transformation via somatic embryogenesis in purslane (Portulaca oleracea L.): an important medicinal plant. DOI 10.1007/s11240-018-1509-3

Comparative study on in vitro micropropagation response of seven globe artichoke [ Cynara cardunculus var. scolymus (L.) Fiori] cultivars: open-pollinated cultivars vs F1 hybrids. DOI: 10.15835/nbha48312004

Omega-3 rich foods: Durum wheat spaghetti fortified with Portulaca oleracea DOI:https://doi.org/10.1016/j.fbio.2020.100730

Antioxidant activity and fatty acids quantification in Sicilian purslane germplasm. NATURAL PRODUCT RESEARCH2020, VOL. 34, NO. 1, 26-33 The quality of functional whole-meal durum wheat spaghetti as affected by inulin polymerization degree https://doi.org/10.1016/j.carbpol.2017.05.081

Germplasm evaluation to obtain inulin with high degree of polymerization in Mediterranean environment NATURAL PRODUCT RESEARCH2020, VOL. 34, NO. 1, 187-191

I suggest to accept the paper after the revisions.

Author Response

Reviewer 2.

Line 43. We did not include a sub tittle Micropropagation asked by the reviewer because in the introduction we defined micropropagation Micropropagation is an in vitro technology of rapidly multiplying elite plants using modern plant tissue culture methods.” Highlighted.

Lines 187 to 191 and 194 to 196: We have included and discussed all the articles suggested and we thank the reviewer on their contribution regarding Cyanara cardunculus var. scolymus (L.) Fiori and on Portulaca oleraceae because more information to the manuscript.

“Sedaghati, B.; Haddad, R.; Bandehpour, M., Efficient plant regeneration and Agrobacterium-mediated transformation via somatic embryogenesis in purslane (Portulaca oleracea L.): an important medicinal plant. Plant Cell Tissue Organ Cult 2019, 136 (2), 231-245.

Padalino, L.; Costa, C.; Conte, A.; Melilli, M. G.; Sillitti, C.; Bognanni, R.; Raccuia, S. A.; Del Nobile, M. A., The quality of functional whole-meal durum wheat spaghetti as affected by inulin polymerization degree. Carbohydr Polym 2017, 173, 84-90.

Ozsan, T.; Onus, A. N., Comparative study on in vitro micropropagation response of seven globe artichoke Cynara cardunculus var. scolymus (L.) Fiori cultivars: open-pollinated cultivars vs F-1 hybrids. Not Bot Horti Agrobot Cluj Napoca 2020, 48 (3), 1210-1220.

Melilli, M. G.; Pagliaro, A.; Scandurra, S.; Gentile, C.; Di Stefano, V., Omega-3 rich foods: Durum wheat spaghetti fortified with Portulaca oleracea. Food Biosci 2020, 37.

Melilli, M. G.; Branca, F.; Sillitti, C.; Scandurra, S.; Calderaro, P.; Di Stefano, V., Germplasm evaluation to obtain inulin with high degree of polymerization in Mediterranean environment. Nat Prod Res 2020, 34 (1), 187-191”.

 And also this one:

 “Govidaraghavan, S.;  Sucher,  N.J. Quality Assessment of medicinal herbs and their extracts: criteria and prerequisites for consistent safety and efficacy of herbal medicine. Epilepsy Behav. 2015, 52:363-371”

Highlighted.

Reviewer 3 Report

The manuscript submitted for evaluation presents an interesting issue concerning micropropagation in the cultivation of plant materials for therapeutic and health-promoting applications. The idea of the article is good, but its realization needs improvement.

The authors presented the topics of herbal raw materials rather one-sidedly, from the perspective of South and North American countries, ignoring the differences in procedures in Asia and Europe. It is known that different countries in the world may have different regulations regarding herbal medicines. In the USA they are treated as dietary supplements, in the EU as certified (standardized) medicinal products, in Asia they have a status of traditional medicines. Medicinal plants for drug production in European countries are subject to different procedures than presented in the manuscript. According to the regulations, they mainly come from crops under supervision. Some plant substances are obtained through biotechnological processes. The European Medicines Agency and its Committee on Herbal Medicinal Products (HMPC), as well as the European Pharmacopoeia, provide guidelines on their quality and purity. I also have reservations about the content of Tables 1 and 2, in which ranges of application are given that are incompatible with those in force, e.g. vincristine and vinblastine are not antihypertension agents. Most of these substances have the status of active pharmaceutical ingredients (API) and are used for strictly defined indications. In addition, the description of medicinal plants should include their full binominal name with the name of the author.

Please note the literature and organize the literature used.

Author Response

Reviewer 3.

Line 108 to 115: We included a sentence on Europe certified medicinal products and Asia as traditional medicine. Most importantly, we added a paragraph on good agricultural, collection, and manufacturing practices and how micropropagation may help on the uniformity and consistency of the products. “In the Europe, the phytomedicines are standardized and certified medicinal products and in Asia they have a status of traditional medicine. Phytomedicinal crops relate to the cultivation of medicinal species by which a mixture of multiple active compounds commercialized as standardized products. Usually, phytomedicines are evaluated for quality as the means to ensure safety, as complex mixtures of secondary compounds, to maintain consistency is fundamental to their efficacy. Thus, authenticity and uniformity and well defined cultivation practices and postharvest processes are essential to certify safety and efficacy. Govidaraghavan and Sucher [44], reinforce that herbal productions must follow good agricultural and collection practice (GACP), good plant authentication and identification practice (GPAIP), good manufacturing practice (GMP) before and during the manufacturing process, and guided by analytical tools, and micropropagation is an important tool ensuring uniformity and consistency in open pollinated crops”. Highlighted.

Regarding Table 1 vinblastine and vincristine are anticancer compounds and not antihypertension, we removed Ajmalicine (antihypertension compound) because it was confusing (Table 1). We also included the authors to the name of the species.

Round 2

Reviewer 1 Report

Corrections are fine, but the title can be made more appropriate. Instead of "Using Micropropagation to Develop Medicinal Plants into Crops", something like  'Micropropogation as a tool for mass cultivation of medicinal and nutraceutical crops' can be tried. This is just a suggestion.

Reviewer 3 Report

AkceptujÄ™ aktualnÄ… wersjÄ™ i nie komentujÄ™.

I accept the current version and do not comment.